# Magnetoelectrical control of nonreciprocal microwave response in a multiferroic helimagnet

Y. Iguchi[1], Y. Nii[1] & Y. Onose[1]

The control of physical properties by external fields is essential in many contemporary technologies. For example, conductance can be controlled by a gate electric field in a field effect transistor, which is a main component of integrated circuits. Optical phenomena induced by an electric field such as electroluminescence and electrochromism are useful for display and other technologies. Control of microwave propagation is also important for future wireless communication technology. Microwave properties in solids are dominated mostly by magnetic excitations, which cannot be easily controlled by an electric field. One solution to this problem is to use magnetically induced ferroelectrics (multiferroics). Here we show that microwave nonreciprocity, that is, different refractive indices for microwaves propagating in opposite directions, could be reversed by an external electric field in a multiferroic helimagnet $Ba_2Mg_2Fe_{12}O_{22}$. This approach offers an avenue for the electrical control of microwave properties.

[1] Department of Basic Science, University of Tokyo, Tokyo 153-8902, Japan. Correspondence and requests for materials should be addressed to Y.I. (email: yiguchi@g.ecc.u-tokyo.ac.jp).

Multiferroics have been investigated extensively since the discovery of magnetically induced ferroelectrics in perovskite manganites[1]. This class of materials exhibits a giant magnetoelectric effect, which is an electric-field-induced change in magnetization, and its reciprocal effect, a magnetic-field-induced change in electric polarization. Magnetoelectric coupling is also valid during the course of magnetic oscillation. A spin wave excitation is coupled to an alternating electric field as well as an alternating magnetic field in multiferroics[2,3]. This dynamical magnetoelectric coupling gives rise to the unique nature of electromagnetic waves. The refractive indices of oppositely propagating electromagnetic waves are different from each other, an effect known as nonreciprocal directional dichroism. Nonreciprocal directional dichroism was first observed for the electronic transition of a non-centrosymmetric molecule in the visible region by Rikken et al.[4] Since then, optical nonreciprocity has been extensively investigated in non-centrosymmetric materials[5–18]. Similar nonreciprocity was also realized for magnon excitation in a transverse conical magnetic state[19–21], which is one of the prototypical spin arrangements exhibiting multiferroic properties, as shown in Fig. 1a. This magnetic state has a net magnetization $M$ in one direction. The magnetic moment components perpendicular to $\mathbf{M}$ rotate along the conical wave vector $\mathbf{q_0}$ perpendicular to $\mathbf{M}$. According to the spin current mechanism[22], the rotating components give rise to static electric polarization $\mathbf{P}$. Thus, $\mathbf{P}$ and $\mathbf{M}$ may oscillate in the spin excitations of the transverse conical state. Miyahara and Furukawa[23] have theoretically shown that the lowest magnon excitation simultaneously induces an alternating $\mathbf{P}$ ($\Delta\mathbf{P}$) and an alternating $\mathbf{M}$ ($\Delta\mathbf{M}$), which is denoted as toroidal magnon. Therefore, this mode is electrically and magnetically active and shows a strong magnetoelectric effect. The resulting large magnetoelectric coupling induces a term $\mathbf{k}\cdot(\mathbf{P}\times\mathbf{M})$ in the refractive index for the electromagnetic wave, indicating a nonreciprocal directional dichroism, which can be controlled by a DC electric field as well as a DC magnetic field. Recently, controllable optical nonreciprocity has been observed for spin excitation in the terahertz region in high magnetic fields of

$3\leq\mu_0 H\leq 7$ T (ref. 19), which is ascribed to the toroidal magnon mode.

Here we report controllable nonreciprocity in ubiquitous frequency (10–15 GHz) and magnetic field (160 mT) ranges in a multiferroic helimagnet $Ba_2Mg_2Fe_{12}O_{22}$ with a longer period and smaller magnetic anisotropy. While microwave nonreciprocity was previously observed in a chiral magnet, it depended on the crystal chirality and could not be reversed by an electric field[13]. In $Ba_2Mg_2Fe_{12}O_{22}$, however, the electric polarization can be controlled by an electric field or a low magnetic field[24]. Related materials show a magnetoelectric response above room temperature[25]. The microwave functionality of this novel material may contribute to the development of interesting future technologies.

## Results

**Variation of magnetic structure in the magnetic field.** Figure 2a shows the crystal structure of $Ba_2Mg_2Fe_{12}O_{22}$. The crystal structure is composed of two types of alternately stacked blocks denoted as S and L blocks. Within each block, the magnetic moments are ferrimagnetically ordered. As a result, the L and S blocks have large and small net magnetic moments, respectively. Figure 2b shows the magnetization curve at 6 K for $Ba_2Mg_2Fe_{12}O_{22}$. At low temperature, the magnetic moments of the S and L blocks showed conical magnetic orderings where the wave vector of the spin structure was $\mathbf{q_0}=(0, 0, 0.59)$[26,27]. Therefore, there was a finite spontaneous magnetization. When the magnetic field was increased, the magnetization curve showed several kinks reflecting magnetostructural transitions, as indicated by inverted triangles in Fig. 2b. While the helical plane was perpendicular to the [001] axis at zero magnetic field, it was inclined and a spontaneous electric polarization was induced parallel to the [1$\bar{1}$0] axis in a small magnetic field along the [110] axis while maintaining $\mathbf{q_0}=(0, 0, 0.59)$. When the magnetic field was further increased, the helical plane became perpendicular to the [110] axis, and a transverse conical state with $\mathbf{q_0}=(0, 0, 3/4)$ emerged at ~60 mT. Then, a transverse conical state with $\mathbf{q_0}=(0, 0, 3/2)$ became dominant above 200 mT (ref. 27). The

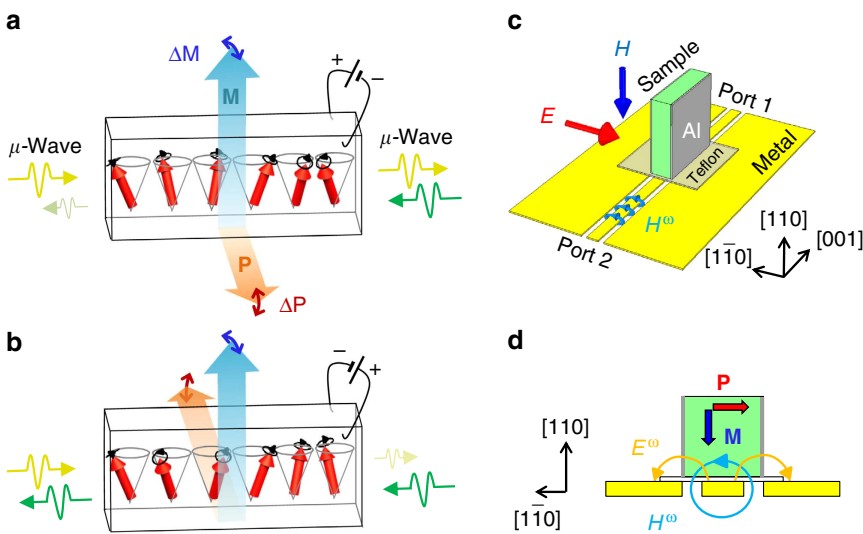

**Figure 1 | Electrical control of nonreciprocal microwave transmission.** (**a,b**) Illustration of nonreciprocal microwave transmission in ferroelectric helimagnets. The strong magnetoelectric coupling of a toroidal magnon in the transverse conical magnetic state provides the nonreciprocity, which is the difference in the refractive indices between oppositely propagating microwaves. The microwave nonreciprocity as well as spin helicity can be controlled by an electric field. (**c**) Experimental set-up of the microwave measurement. (**d**) Cross-sectional view of the (001) plane and the alternating magnetic field $H^\omega$ and electric field $E^\omega$ of microwaves in the experimental set-up.

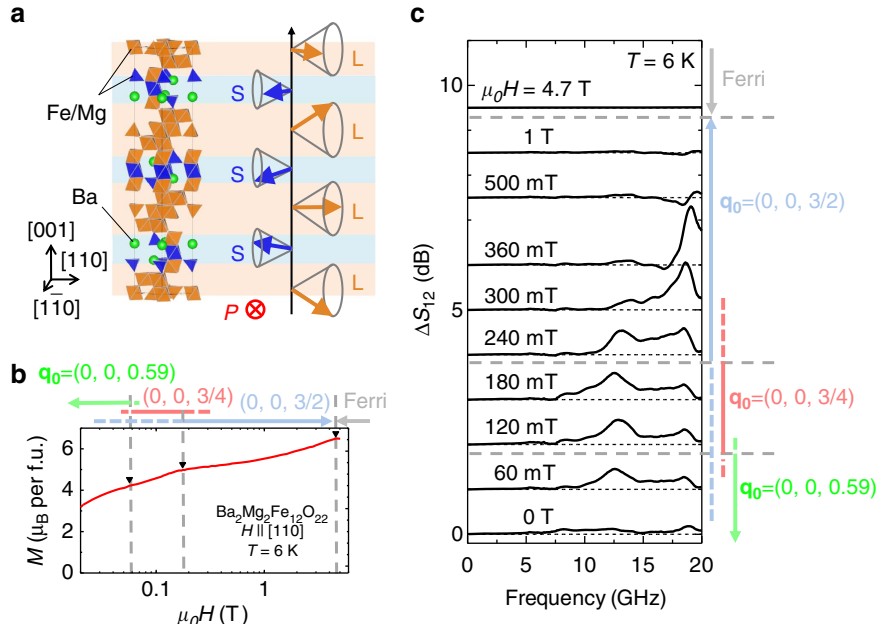

**Figure 2 | Crystal structure and magnetic properties of Ba$_2$Mg$_2$Fe$_{12}$O$_{22}$.** (**a**) Crystal structure of Ba$_2$Mg$_2$Fe$_{12}$O$_{22}$ and spin structure in the **q$_0$** = (0, 0, 3/4) state, in which the control of microwave nonreciprocity is demonstrated. (**b**) Magnetization curve of Ba$_2$Mg$_2$Fe$_{12}$O$_{22}$ at 6 K in a magnetic field parallel to the [110] direction. (**c**) Microwave absorption $\Delta S_{12}$ in several magnetic fields parallel to the [110] direction at 6 K.

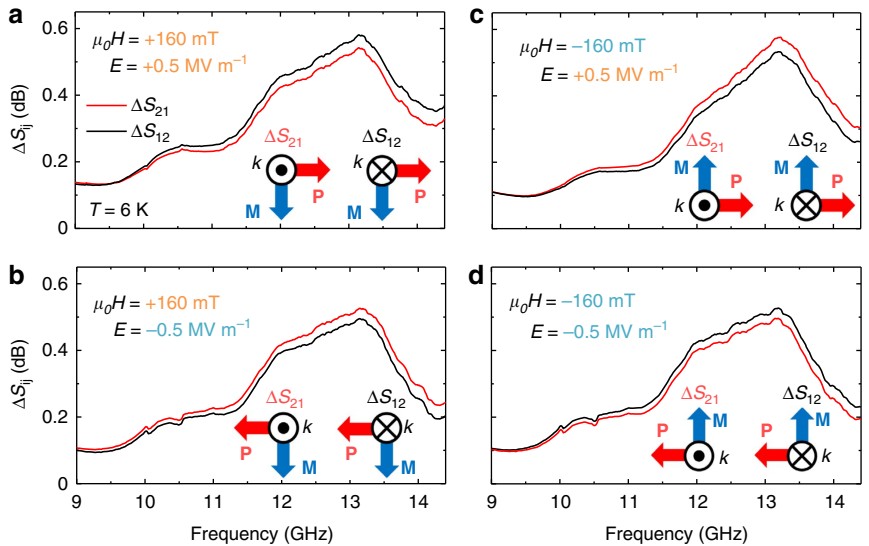

**Figure 3 | Magnetoelectrical control of microwave nonreciprocity.** (**a–d**) Microwave absorption $\Delta S_{12}$, $\Delta S_{21}$ at 6 K in ±160 mT (+160mT in (**a,b**) and −160 mT in (**c,d**)). The poling fields were $E = \pm 0.5\,\mathrm{MV\,m^{-1}}$ (+0.5 MVm$^{-1}$ in (**a,c**) and −0.5 MVm$^{-1}$ in (**b,d**)). Inset illustrates the directional relation of $P$, $M$ and $k$ for $\Delta S_{12}$ and $\Delta S_{21}$ measurements.

magnetic structure in the **q$_0$** = (0, 0, 3/4) state is shown in Fig. 2a. The phase transitions between helimagnetic structures were first-order transitions, and phase coexistence was observed around the phase boundaries. In particular, the **q$_0$** = (0, 0, 3/2) state extended down to around zero magnetic field. The **q$_0$** = (0, 0, 3/2) state showed a subtle spin structural change at ∼200 mT (ref. 27). Above 4.5 T, a collinear ferrimagnetic state appeared, and the ferroelectricity was quenched.

**Microwave absorption.** Figure 2c shows $\Delta S_{12}$ spectra at various magnetic fields at 6 K, measured using the experimental set-up shown in Fig. 1c. The spatial distribution of electric and magnetic

fields of microwaves is shown in Fig. 1d. The $\Delta S_{12}$ spectra reflect the absorption of microwaves owing to magnetic resonance during the course of propagation from port 2 to port 1. For details of the experimental set-up and a precise definition of $\Delta S_{12}$, see Methods. While the microwave absorption was almost absent at zero field, a small and broad peak appeared at ∼12 GHz when the magnetic field was applied along the [110] direction. When the magnetic field was increased above the phase boundary between the **q$_0$** = (0, 0, 0.59) and **q$_0$** = (0, 0, 3/4) states (≈60 mT), the frequency was slightly increased, and the intensity also increased (for a detailed comparison of frequency, see also Supplementary Fig. 1a,b). The peak at ∼12 GHz was suppressed, and the high-frequency peak at ∼19 GHz evolved instead near

the phase boundary between the $\mathbf{q_0} = (0, 0, 3/4)$ and $\mathbf{q_0} = (0, 0, 3/2)$ states ($\approx 200$ mT). Two peaks were simultaneously observed around the phase boundary because of the phase coexistence. The higher peak frequency increased with magnetic field and disappeared from the measured frequency range above 500 mT. The high-frequency peak was suppressed below 200 m. However, it did not vanish but coexisted with the lower frequency peak even around zero field, which was consistent with the presence of the $\mathbf{q_0} = (0, 0, 3/2)$ state around zero field. We demonstrated the control of nonreciprocity mainly for the low-frequency peak in the $\mathbf{q_0} = (0, 0, 3/4)$ state because the measurement sensitivity in the frequency range of 9–14 GHz was better than that in the higher-frequency region in our experimental set-up.

**Magnetoelectrical control of nonreciprocity.** Figure 3 demonstrates the magnetoelectrical control of nonreciprocal microwave absorption. $\Delta S_{21}$ is a microwave absorption spectrum similar to $\Delta S_{12}$, but with the opposite microwave propagation direction. We performed a poling procedure using an external electric field in order to fix the spin helicity (for the details, see Methods). After the poling procedure, we turned off the external electric field and changed the magnetic field to a certain value and then measured $\Delta S_{12}$ and $\Delta S_{21}$. Figure 3a,c shows the spectra measured after the poling procedure with an electric field $E$ of $+0.5$ MV m$^{-1}$, and Fig. 3b,d shows those measured with $E = -0.5$ MV m$^{-1}$. The measured magnetic field $H$ was $+160$ mT for Fig. 3a,b and $-160$ mT for Fig. 3c,d. For all cases, there were clear differences between $\Delta S_{12}$ and $\Delta S_{21}$, indicating the microwave nonreciprocity. The nonreciprocity was reversed by the inversion of either $E$ or $H$, but it was unchanged by the simultaneous inversion of $E$ and $H$.

To further study the effect of an external field, we investigated the poling electric field dependence of microwave nonreciprocity, $\Delta S_{12} - \Delta S_{21}$, at $\mu_0 H = 160$ mT and the magnetic field dependence at $E = \pm 0.5$ MV m$^{-1}$. The results are shown in Fig. 4a,b, respectively. The sign of the nonreciprocity reflected that of $E$, and the magnitude monotonically increased with $E$. The frequency of the nonreciprocity increased with the magnetic field, corresponding to the increase of absorption peak frequency (see also Supplementary Fig. 2). In Fig. 4c, we show the integrated intensity of nonreciprocity $I_{12}$ between 9 and 14.4 GHz at $\mu_0 H = \pm 160$ mT as a function of $E$. The sign of $I_{12}$ depended on those of $E$ and $H$, and the magnitude of $I_{12}$ gradually increased in the low-$E$ region and tended to be saturated at $\sim 0.5$ MV m$^{-1}$. A similar $E$ dependence was discerned in the polarization results. Therefore, these field dependences were dominated by the ferroelectric domain population.

## Discussion

We observed controllable nonreciprocity for the lowest-energy magnetic resonance modes in the transverse conical state. The observed nonreciprocity cannot be ascribed to the effect of magnetic dipole interaction because it should not be changed by the poling electric field in the case of magnetic dipolar nonreciprocity. As explained in the Introduction, the lowest-energy magnetic resonance in the transverse conical state is the toroidal magnon mode, which is expected to show a large nonreciprocity along $\mathbf{P} \times \mathbf{M}$. This is quite consistent with the present observations. While similar magnon modes were previously observed at large magnetic fields in the THz region for perovskite $R$MnO$_3$ (refs 19,20), we have observed it here at a low magnetic field in the GHz region. While $R$MnO$_3$ and Ba$_2$Mg$_2$Fe$_{12}$O$_{22}$ are both helimagnets, the period and magnetic anisotropy are different. For TbMnO$_3$, which is a typical material of perovskite helimagnets, the period is 2 nm, and the magnetic anisotropy constant $K_1$ is $6 \times 10^7$ erg cm$^{-3}$, whereas for Ba$_2$Mg$_2$Fe$_{12}$O$_{22}$, the period is 23 nm and the magnetic anisotropy constant $K_1 + 2K_2$ is $-6 \times 10^5$ erg cm$^{-3}$ (refs 28–31). The frequency difference reflects the differences in the magnetic anisotropy[32] and helical period. The magnitude of nonreciprocity in this study was as large as 6–8%, which is smaller than that observed in the THz region. One of the reasons for this is that the intensity of pure magnetic excitation becomes relatively large and, therefore, the relative nonreciprocity becomes small in the case of the spontaneous conical state. One of the advantages of microwave nonreciprocity is compatibility with other microwave technologies. For example, the nonreciprocity can be adequately enhanced by utilizing a high-Q resonator. While the electrical control of microwave properties has been extensively investigated for multiferroic heterostructures mainly with the use of mechanical strain-mediated magnetoelectric coupling[33], controllable nonreciprocity in the transverse conical state seems more useful. Thus, the new microwave functionality of controllable nonreciprocity demonstrated here has a large potential for practical applications.

## Methods

**Sample preparation and magnetization measurement.** A single crystal of Ba$_2$Mg$_2$Fe$_{12}$O$_{22}$ was grown by the flux method[26]. The crystal orientations were determined by X-ray diffraction. The magnetization curve was measured using a superconducting quantum interference device magnetometer (Magnetic Property Measurement System, Quantum Design) at 6 K in magnetic fields parallel to the [110] direction.

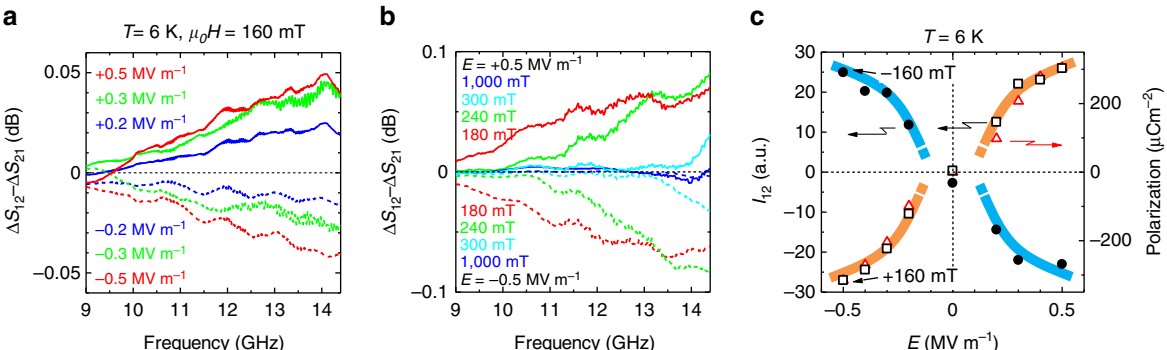

**Figure 4 | Electric field and magnetic field dependences of microwave nonreciprocity. (a)** Nonreciprocities, $\Delta S_{12} - \Delta S_{21}$, at $+160$ mT measured after poling at various electric field strengths. **(b)** Nonreciprocities at several magnetic fields measured after electric field poling $E = +0.5$ MV m$^{-1}$ (solid lines) and $E = -0.5$ MV m$^{-1}$ (dashed lines). **(c)** Integrated intensities of nonreciprocity $I_{12}$ at $+160$ mT (open squares) and $-160$ mT (closed circles) are plotted as a function of the poling electric field $E$. The polarization at $+160$ mT (open triangle) is also plotted as a function of $E$ for comparison. The solid lines are merely guides for the eyes.

**Polarization measurement and poling procedure.** The spontaneous electric polarization along the [1$\bar{1}$0] direction at 6 K was obtained by the integration of measured displacement currents. Before the current measurement, we performed the following poling procedure to fix the spin helicity in a manner similar to a previous study[24]. After cooling the sample to 50 K without external fields, an electric field was applied in the [1$\bar{1}$0] direction, and then a magnetic field as large as 5 T was applied parallel to the [110] direction. Next, the magnetic field was decreased to 1 T, followed by cooling to 6 K. Finally, the electric field was removed. The displacement current was measured while sweeping the magnetic field.

**Microwave spectroscopy.** We fabricated a microwave device composed of a $Ba_2Mg_2Fe_{12}O_{22}$ sample and a microwave coplanar waveguide shown in Fig. 1c in order to measure the microwave response after the electric poling procedure. The dimensions of the sample were 1.1 mm × 1.1 mm × 0.3 mm. The largest plane was perpendicular to the [1$\bar{1}$0] direction, and the two longer sides were parallel to the [110] and [001] directions. The aluminium electrodes were attached to the largest sample planes for applying the electric field. The coplanar waveguide was designed so that the characteristic impedance matched 50 Ω. The width of the strip line was 0.2 mm, and the gap between the strip line and the ground plane was 0.05 mm. The sample was placed on the coplanar waveguide so that the [001] direction was parallel to the wave guide. A Teflon sheet with a thickness of 20 μm was inserted between the sample and the waveguide for insulation. The microwave response was measured in a superconducting magnet using a vector network analyser (E5071C, Agilent). Before the microwave measurement, we performed a poling procedure similar to that used previously for the polarization measurement. For the measurements in negative magnetic fields, the magnetic field during the poling was also negative. The microwave absorption spectrum $\Delta S_{12}$ at $\mu_0 H$ is defined as $|S_{12}(5\,T)| - |S_{12}(\mu_0 H)|$, where $S_{12}$ is the microwave transmittance from port 2 to port 1. Because the magnetic resonance frequency was high enough in the entire measurement range at 5 T, the microwave absorption owing to magnetic resonance can be obtained using this formula. The microwave absorption with the opposite wave vector $\Delta S_{21}$ was also obtained by a similar procedure.

**Data availability.** All relevant data are available from the corresponding authors upon request.

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

## Acknowledgements

We thank S. Hirose for constructive discussion. This work was supported in part by Grants-in-Aid for Scientific Research (Grant Nos 25247058, 16H04008 and 15K21622). Y.I. is supported by the Grant-in-Aid for Research Fellowship for Young Scientists from the Japan Society for the Promotion of Science (No. 16J10076).

## Author contributions

Y.I. carried out the crystal growth, pyroelectric current measurement, coplanar wave-guide preparation and microwave measurement and analysed the data. Y.N. contributed to the modification of the microwave experimental set-up for applying the electric field. Y.O. supervised the project. Y.I. wrote the paper based on discussion with and assistance from Y.N. and Y.O.

## Additional information

**Competing interests:** The authors declare no competing financial interests.

**Publisher's note**: 

