## [Peer Review File · Nature Communications]

Reviewers' comments:

Reviewer #1 (Remarks to the Author):

This manuscript presents an intriguing result of the magnetoelectrical control of nonreciprocal microwave response with a multiferroic helimagnet Ba₂Mn₂Fe₁₂O₂₂. The nonreciprocal directional dichroism (NRD), represents the difference between the oppositely propagating electromagnetic waves, which has become a active field of research due to its potential applications. Although the previous works has demonstrated both the magnetic field and magnetoelectric coupling of the NRD at THz, the current study extends the approach to GHz with a low magnetic field control of NRD in the model system of multiferroic helimagnet. This interesting result will provoke wide range of research interests in the field of multiferroic. I would like to recommend the paper to be published in nature communications with a few suggestions.

1) Several highly relevant references are missing from the current paper, such as: Nature communications 5, 4583 (2014) and Phys. Scr. 81, 065703 (2010). Phys. Rev. Lett. 106, 057403 (2011), Nature Phys. 8, 734 (2012), PRL 101, 117402 (2008), PRL 115, 127203 (2015), PRL 115, 267207 (2015), Nature Communications 5, 5203 (2014), etc. The reviewer suggests the authors to carry out a more careful reference survey with suitable change in the introduction and discussion parts.

2) The author used the microwave absorption spectra measured at 5 Tesla as the reference to calculate the absorption spectra at other fields. However, it would be clear to present the NRD with fixed magnetic field and opposite microwave propagation, unless the author can give a solid proof that the high magnetic field PE phase dose not have NRD effect.

3) A detailed discussion about the NRD effect observed in the current system and other systems, such as the RMnO₃, would be necessary for the general readers to understand the intrinsic different between the THz and GHz regions.

Reviewer #2 (Remarks to the Author):

The authors report the observation of non-reciprocal propagation of magnons with frequencies around 10 GHz in the multiferroic hexaferrite Ba₂Mg₂Fe₁₂O₂₂. They show that the sign of the non-reciprocal effect can be reversed by reversing the direction of either the electric or magnetic field. The effect is observed in the conical spiral state of the hexaferrite, which has a net magnetic moment controlled by the applied magnetic field and an electric polarization induced by the spiral component, which is controlled by the applied electric field. The frequency 10-15 GHz is apparently the frequency of the excitation mode of the conical spiral corresponding to the precession of the magnetic moment and tilts of the spiral plane.

This is a nice piece of work, but the I do not find these result novel enough to justify publication in the Nature Communications.

The non-reciprocity in the microwave range was observed previously: in addition to Ref.[7], it was reported by Y. Iguchi, Y. Onose et al, Nonreciprocal magnon propagation in a noncentrosymmetric ferromagnet LiFe₅O₈, PRB 92, 184419 (2015) (for some reasons, the authors do not cite their own paper) as well as in S. Seki et al. Phys. Rev. B 93, 235131 (2016).

These microwave non-reciprocity was observed in chiral magnets with inversion symmetry broken by the crystal lattice, in which case the sign of the non-reciprocal effect can only be reversed magnetically. The electric reversal was achieved in another multiferroic material, albeit in the THz frequency range (Ref. [6]).

The novelty of the work described in the manuscript is thus in the electric reversal of non-reciprocity at microwave frequencies, which does not look to me as a big step forward, especially, since the electric switching demonstrated in this work and in Ref. [6] is not very useful for device applications. Namely, the electric field was applied on cooling the sample from 50 K to 6 K in order to select a single ferroelectric domain, as described in Methods. The measurements were done at zero electric field. To reverse the polarization direction, the sample had to be heated and cooled again, which is an extremely slow process.

In addition, the manuscript is not very carefully written. For example, what the microwave non-reciprocity means is only explained in the Fig. 1 caption.

On page 2 it is stated that the non-reciprocal effect is observed at the frequency of one of the Nambu-Goldstone mode, which makes a reader wonder which mode it is. The magnetic excitation in question cannot be a Nambu-Goldstone boson, as it has a finite frequency. The Nambu-Goldstone boson is the phason (the translation of the spiral component along the wave vector) and it is not coupled to the electric field. The mode that gives rise to the observed effect corresponds to periodic tilts of the spiral plane. There is no symmetry breaking associated with this mode, because the orientation of the spiral plane is determined by the applied magnetic field and magnetic anisotropy of the hexaferrite.

Furthermore, English requires improvement, e.g.:

"This magnetic state has net magnetization M along one direction."

"..., which is denoted as nonreciprocal directional dichroism."

"Thus P as well as M may be oscillated ..."

"ferroelectricity is induced parallel to the $[1-10]$ axis"

In view of this criticism, I cannot recommend publication of this manuscript in the Nature Communications. My recommendation to the authors is to submit their manuscript to a more specialized journal.

Reviewer #3 (Remarks to the Author):

Nonreciprocal optical effects are in the focus of recent solid-state research in the field of multiferroics and finally, from an application perspective, will enable the optical control of light. This directional dichroism, the difference in forward and backward propagation of light through a material, usually is small effect. So far, optical non-reciprocity has been studied at relatively high external magnetic fields in the THz regime. Enhanced directional dichroism has been observed utilizing the electromagnon response in multiferroics. In the recent work by Iguchi et al., microwave non-reciprocity was detected in the multiferroic helimagnet $\text{Ba}_2\text{Mg}_2\text{Fe}_{12}\text{O}_{22}$ in the external magnetic field range of some hundred mT.

To study microwave absorption in external electric and magnetic fields, Iguchi et al. have utilized coplanar wave-guide spectroscopy, where a millimeter-sized sample is placed on a microwave device. The authors have conducted detailed and systematic studies as function of external magnetic and electric fields. The results as documented in Figs 3 and 4, although the effects of non-reciprocity are small, are convincing and show directional dichroism as function of switching electric and magnetic fields.

The data presented in this manuscript certainly are technically sound and the results are novel. The main question is, if this manuscript does contain important advances of significance to specialists within the field of magnetoelectric effects. Despite the fact that similar results have been published earlier, e.g., at THz frequencies at high fields (Kezsmarki et al., Phys. Rev. Lett. 106, 057403, 2011; Takahashi et al.; Ref. 6 of this manuscript), or in Skyrmion crystals, where the nonreciprocal optical effect cannot be reversed by external magnetic fields (Okamura et al., Ref. 7 of this manuscript), I think that this publication contains enough novelty and significance to experts in the field to justify publication in Nature Communications: These experiments have been performed on a new multiferroic compound at microwave frequencies (5 - 20 GHz) and the effect has been documented at very low external magnetic fields (< 300 mT).

Some remarks and suggestions:

A closer inspection of Fig. 2c indicates that the phase boundaries between the different multiferroic phases are not well defined. The complexity of the phase boundaries also is documented in Ref. 11. It is unclear if the phase boundaries are of 1st or 2nd order. I think this has to be described in more detail. I would suggest that effects of non-reciprocity between 10 and 15 GHz exists only in the FE2 phase. No finite difference between forward and backward propagation is visible in zero external magnetic field. Finite intensity at 60 mT could result from ill-defined phase boundaries, precursor or fluctuation effects or from finite hysteresis. In addition, the strong increase in the difference spectra for magnetic fields > 300 mT obviously is the characteristics of phase FE3. It is unclear if the small difference between 0 and 200 mT at high frequencies (> 14 GHz) has to be taken serious, or can be explained by experimental uncertainties. This has to be discussed in more detail in the revised version of the manuscript.

In Fig. 2c, it certainly would be helpful to include a spectrum at higher magnetic fields (> 4 T) taken in the paraelectric phase, to document experimental uncertainties.

The discussion of Fig. 2b is misleading. Even at very low magnetic fields, the magnetization M exceeds already $3 \mu\text{B}$, which is almost half of the full magnetization. This is significant and by no means "small but finite".

I think that the work by Kezsmarki et al., Phys. Rev. Lett. 106, 057403, 2011, has to be cited.

To summarize, after taking the above suggestion into account, I think that the manuscript can be published in Nat. Commun.

REVIEWERS' COMMENTS:

Reviewer #2 (Remarks to the Author):

I have read the authors' reply to my and other referees' questions. I find the answers satisfactory. The findings described in the manuscript are interesting and the experimental work looks solid. The clarity of the presentation and English can still be improved, but I leave it to Editors. I recommend this manuscript for publication.

Reviewer #3 (Remarks to the Author):

In the revised version of the manuscript, the authors followed most of my suggestions and answered most of the questions raised in my report. Also after reading the comments of the other referees, I still think that this manuscript contains enough novelty and significance to be published in Nature Communications.

Authors' response to the reviewer #1's comments

First of all, we thank the reviewer for valuable comments.

Followings are our straightforward responses to them.

Comment (1): Several highly relevant references are missing from the current paper, such as: Nature communications 5, 4583 (2014) and Phys. Scr. 81, 065703 (2010). Phys. Rev. Lett. 106, 057403 (2011), Nature Phys. 8, 734 (2012), PRL 101, 117402 (2008), PRL 115, 127203 (2015), PRL 115, 267207 (2015), Nature Communications 5, 5203 (2014), etc. The reviewer suggests the authors to carry out a more careful reference survey with suitable change in the introduction and discussion parts.

Response (1): Following the recommendation we have carried out a more careful reference survey and added a number of references including those suggested by the reviewer (Ref. [4]-[12],[14]-[18]).

Comment (2): The author used the microwave absorption spectra measured at 5 Tesla as the reference to calculate the absorption spectra at other fields. However, it would be clear to present the NRD with fixed magnetic field and opposite microwave propagation, unless the author can give a solid proof that the high magnetic field PE phase does not have NRD effect.

Response (2): In our experimental setup there is a small but finite difference between S12 and S21 even without any magnetic sample, presumably due to imperfect calibration or circuit nonlinearity. The difference is not dependent on the sign and magnitude of the magnetic field. Therefore the difference can be clearly discriminated from the nonreciprocity caused by the sample. To exclude the magnetic field independent offset we have to subtract the reference 5 T data from the other data. We show the detail magnetic field dependence of nonreciprocity in Fig. S2 in Supplementary Information. It is clear from this figure that the nonreciprocity is negligible above 500 mT. This is the solid proof of absence of NRD effect in PE phase.

Comment (3): A detailed discussion about the NRD effect observed in the current system and other systems, such as the RMnO₃, would be necessary for the general readers to understand the intrinsic difference between the THz and GHz regions.

Response (3): While RMnO₃ and Ba₂Mg₂Fe₁₂O₂₂ are both helimagnets, the period and magnetic anisotropy are quite different. For TbMnO₃, which is the representative material of perovskite helimagnets, the period is 2 nm (Ref. [28]) and magnetic anisotropy constant K_1 is 6×10^7 erg/cm³ (Ref. [29]). On the other hand for Ba₂Mg₂Fe₁₂O₂₂ the period is 23 nm (Ref. [30]) and magnetic anisotropy constant $K_1 + 2 \times K_2$ is -6×10^5 erg/cm³ (Ref. [31]). The frequency difference reflects them. To address this point we added sentences to the Discussion chapter.

All the changes made are listed below:

1. Following the reviewer 1's comment (1) and the reviewer 3's comment (4), we have newly cited references and added sentences in the introduction (page 2, 1st paragraph, 10th-12th line "Nonreciprocal directional... materials [5]-[18]")
2. To respond to the reviewer 1's comment (2), we have added the Fig. S2 in the Supplementary Information.
3. To respond to the reviewer 1's comment (3), we have added sentences to the Discussion chapter (page 5, 3rd paragraph, 9th - page 6, 1st paragraph, 1st line "While RMnO₃ and ...and helical period.").
4. To respond to the reviewer 2's comment (4), we have omitted the words "Nambu-Goldstone mode", and alternatively used the words "toroidal magnon mode" in the main text.

5. To respond to the reviewer 2's comment (5), we have improved the English in the revised manuscript by using a commercial proofreading service.
6. To respond to the reviewer 3's comment (1), we revised the first section ("Variation of magnetic structure in magnetic field") and second section ("Microwave absorption") in the Results chapter and newly added the Fig. 4b in the main text and Fig. S1 and S2 in the Supplementary Information.
7. Following the reviewer 3's comment (2), we newly added the absorption spectra in paraelectric (Ferri) phase to Fig. 2(c) and Fig. S2.
8. Following the reviewer 3's comment (3), we have modified the expression about the magnitude of magnetization (page 3, second paragraph, 7th-8th line "Therefore there was a finite spontaneous magnetization.").

Authors' response to the reviewer #2's comments

First of all, we thank the reviewer for valuable comments.

Followings are our straightforward responses to them.

Comment (1): The non-reciprocity in the microwave range was observed previously: in addition to Ref.[7], it was reported by Y. Iguchi, Y. Onose et al, Nonreciprocal magnon propagation in a noncentrosymmetric ferromagnet LiFe₅O₈, PRB 92, 184419 (2015) (for some reasons, the authors do not cite their own paper) as well as in S. Seki et al. Phys. Rev. B 93, 235131 (2016).

Response (1): The above papers study the nonreciprocal propagation of magnons, which is conceptually different from the nonreciprocal propagation of electromagnetic waves around the magnon excitation frequency studied in the present paper. That is why we did not cite these papers.

Comment (2): The novelty of the work described in the manuscript is thus in the electric reversal of non-reciprocity at microwave frequencies, which does not look to me as a big step forward, especially, since the electric switching demonstrated in this work and in Ref. [6] is not very useful for device applications. Namely, the electric field was applied on cooling the sample from 50 K to 6 K in order to select a single ferroelectric domain, as described in Methods. The measurements were done at zero electric field. To reverse the polarization direction, the sample had to be heated and cooled again, which is an extremely slow process.

Response (2): The sign of microwave nonreciprocity is governed by that of the electric polarization. In this sense the control of the microwave nonreciprocity is equivalent to that of electric polarization. The polarization can be controlled by the external electric field but large magnitude of electric field is needed deep inside ferroelectric phases. In this case the maximum electric field is limited by the presence of quite small number of residual carriers. Electric break down is observed above $E = 0.6$ MV/m. We adapt an alternative way of polarization (equivalently microwave nonreciprocity) control; we applied the external electric field when ferroelectric-paraelectric transition happens because the coercive field becomes minimal at the transition point. For the detail procedure we follow Ishiwata et al. (Ref. [24]). As the reviewer suggested this procedure includes thermal process and takes long time, which cannot be directly used for device applications. Nevertheless we would like to emphasize that our result shows the electrical switching without thermal procedure is in principle possible. After resolving the residual carrier problem the sign of microwave nonreciprocity can be reversed only by the electrical field. In fact the electrical field switching of magnetization that is coupled to the electric polarization has been observed in related materials, which are annealed at high temperature in O₂ gas to reduce the residual carriers and to increase the resistivity (Y.S.Chai et al., Nat. Commun. 2014, and S. Shen et al., Sci. Rep. 2015).

Comment (3): In addition, the manuscript is not very carefully written. For

example, what the microwave non-reciprocity means is only explained in the Fig. 1 caption.

Response (3): Reviewer 2's suggestion is not correct. We have also explained the microwave nonreciprocity in abstract, 9th-10th line "microwave nonreciprocity, which is difference between oppositely propagating microwaves". (In the revised manuscript, the sentence was slightly modified following the suggestion of the proofreading service.)

Comment (4): On page 2 it is stated that the non-reciprocal effect is observed at the frequency of one of the Nambu-Goldstone mode, which makes a reader wonder which mode it is. The magnetic excitation in question cannot be a Nambu-Goldstone boson, as it has a finite frequency. The Nambu-Goldstone boson is the phason (the translation of the spiral component along the wave vector) and it is not coupled to the electric field. The mode that gives rise to the observed effect corresponds to periodic tilts of the spiral plane. There is no symmetry breaking associated with this mode, because the orientation of the spiral plane is determined by the applied magnetic field and magnetic anisotropy of the hexaferrite.

Response (4): In a helical magnet without magnetic anisotropy, there are three zero energy magnon modes at zero magnetic field: $q=0$ mode (phason mode), $+q_0$ mode, and $-q_0$ mode (see, for example, supplementary material of ref. 23). These three modes are definitely Nambu-Goldstone modes in the helimagnetic state. When the external magnetic field is applied, the frequencies of $+q_0$ and $-q_0$ modes becomes finite frequency and show the large nonreciprocity while that of $q=0$ mode remains zero. The observed magnon mode seems corresponding to the $q=q_0$ mode (the lower frequency mode of $+q_0$ mode and $-q_0$ mode). As the reviewer suggested, some readers may be confused by the usage of the terminology "Nambu-Goldstone mode" for the $q=q_0$ mode in the conical magnetic state. The authors of ref. 23 called the $q=q_0$ magnon "topoidal magnon" because this mode oscillates the electric and magnetic dipoles, simultaneously. In the revised manuscript, we used the terminology, alternatively.

Comment (5): Furthermore, English requires improvement, e.g.:
"This magnetic state has net magnetization M along one direction."
"..., which is denoted as nonreciprocal directional dichroism."
"Thus P as well as M may be oscillated ..."
"ferroelectricity is induced parallel to the $[1-10]$ axis"

Response (5): We have tried to improve the manuscript by using a commercial proofreading service.

All the changes made are listed below:

1. Following the reviewer 1's comment (1) and the reviewer 3's comment (4), we have newly cited references and added sentences in the introduction (page 2, 1st paragraph, 10th-12th line "Nonreciprocal directional... materials [5]-[18]")
2. To respond to the reviewer 1's comment (2), we have added the Fig. S2 in the Supplementary Information.
3. To respond to the reviewer 1's comment (3), we have added sentences to the Discussion chapter (page 5, 3rd paragraph, 9th - page 6, 1st paragraph, 1st line "While RMnO_3 and ...and helical period.").
4. To respond to the reviewer 2's comment (4), we have omitted the words "Nambu-Goldstone mode", and alternatively used the words "toroidal magnon mode" in the main text.
5. To respond to the reviewer 2's comment (5), we have improved the English in the revised manuscript by using a commercial proofreading service.
6. To respond to the reviewer 3's comment (1), we revised the first section ("Variation of magnetic structure in magnetic field") and second section ("Microwave absorption") in the Results chapter and newly added the Fig.

4b in the main text and Fig. S1 and S2 in the Supplementary Information.

7. Following the reviewer 3's comment (2), we newly added the absorption spectra in paraelectric (Ferri) phase to Fig. 2(c) and Fig. S2.
8. Following the reviewer 3's comment (3), we have modified the expression about the magnitude of magnetization (page 3, second paragraph, 7th-8th line "Therefore there was a finite spontaneous magnetization.").

Authors' response to the reviewer #3's comments

First of all, we thank the reviewer for valuable comments.

Followings are our straightforward responses to them.

Comment (1): A closer inspection of Fig. 2c indicates that the phase boundaries between the different multiferroic phases are not well defined. The complexity of the phase boundaries also is documented in Ref. 11. It is unclear if the phase boundaries are of 1st or 2nd order. I think this has to be described in more detail. I would suggest that effects of non-reciprocity between 10 and 15 GHz exists only in the FE2 phase. No finite difference between forward and backward propagation is visible in zero external magnetic field. Finite intensity at 60 mT could result from ill-defined phase boundaries, precursor or fluctuation effects or from finite hysteresis. In addition, the strong increase in the difference spectra for magnetic fields > 300 mT obviously is the characteristics of phase FE3. It is unclear if the small difference between 0 and 200 mT at high frequencies (> 14 GHz) has to be taken serious, or can be explained by experimental uncertainties. This has to be discussed in more detail in the revised version of the manuscript.

Response (1): As the reviewer suggested, the explanation of multiferroic phases was not clear. According to the neutron diffraction experiment (Ref. [27]), there are three magnetic states with different wave vectors. The lowest

field phase is the longitudinal conical state with incommensurate wave vector $q_0 = (0, 0, 0.59)$. The second and the third states are transverse conical states with $q_0 = (0, 0, 3/4)$ and $q_0 = (0, 0, 3/2)$. In the previous manuscript, we called these states FE1, FE2, and FE3. We have found that the abbreviations were similar to but slightly different from those used in ref. 27. In order to avoid the confusion, we call these states by the wave vectors in the revised manuscript. The transitions between helimagnetic phases with different q_0 are 1st order and the phase coexistence is clearly observed around the phase boundaries. In particular, the $q_0 = (0, 0, 3/2)$ state was extended down to around zero magnetic field and coexists with the low field states. Precisely speaking, in ref. [27], a subtle spin structural change was observed for the $q_0 = (0, 0, 3/2)$ state. Above around 200mT, the magnetic moment of L block is along to the (001) plane while that of S-block below around 200mT. In the revised manuscript, we briefly noted this spin structural change.

In the present microwave experiment two broad peaks are observed in the absorption spectra. One is discerned around 10-15 GHz and the other is around 19 GHz. The detail magnetic field dependence of microwave absorption is newly shown in the Supplementary Information. The lower frequency peak is almost absent at zero field but observed at 20 mT. Around the boundary between $q_0 = (0, 0, 0.59)$ and $(0, 0, 3/4)$ (around 60 mT), the peak frequency shows a slight but discernible change. Therefore, the absorption below 60 mT is not induced by the simple phase coexistence but both the states seem to show absorption in this frequency range. Above 360 mT the lower frequency peak disappears. On the other hand the higher frequency peak is observed mainly above 200 mT and the frequency increases with magnetic field. The origin of high frequency peak seems magnon excitations in the $q_0 = (0, 0, 3/2)$ phase.

The peak coexistence between 180-300 mT is induced by the phase coexistence of $q_0 = (0, 0, 3/4)$ and $q_0 = (0, 0, 3/2)$ phases. The small trace of high frequency peak below 180 mT may also be ascribed to the phase coexistence, because the $q_0 = (0, 0, 3/2)$ state is observed even around zero magnetic field in the neutron experiment(Ref. [27]). The data in the revised manuscript are newly obtained by the additional experiment after the 1st submission. The small trace is also reproduced.

The microwave nonreciprocity was clearly observed above 60 mT in both the $q_0=(0,0,3/4)$ and $q_0=(0,0,3/2)$ phases as shown in Supplementary Information.

To address these points, we added figures about the detailed magnetic field dependences of microwave absorption and nonreciprocity to the Supplementary Information (Fig. S1 and S2) and the main text (Fig. 4b), and we also added the description about the phase coexistence and interpretation of absorption peaks to the main text.

Comment (2): In Fig. 2c, it certainly would be helpful to include a spectrum at higher magnetic fields (> 4 T) taken in the paraelectric phase, to document experimental uncertainties.

Response (2): Following the reviewer's comment we newly show the absorption spectra of PE phase in Fig. 2(c) and Fig. S2.

Comment (3): The discussion of Fig. 2b is misleading. Even at very low magnetic fields, the magnetization M exceeds already $3 \mu\text{B}$, which is almost half of the full magnetization. This is significant and by no means "small but finite".

Response (3): We have corrected the expression about the magnitude of magnetization (page 3, first paragraph, 7th line "Therefore there is a finite spontaneous magnetization.").

Comment (4): I think that the work by Kezsmarki et al., Phys. Rev. Lett. 106, 057403, 2011, has to be cited.

Response (4): We added the citation to the reference (Ref. [11]).

All the changes made are listed below:

1. Following the reviewer 1's comment (1) and the reviewer 3's comment (4), we have newly cited references and added sentences in the introduction (page 2, 1st paragraph, 10th-12th line "Nonreciprocal directional... materials [5]-[18]")
2. To respond to the reviewer 1's comment (2), we have added the Fig. S2 in the Supplementary Information.
3. To respond to the reviewer 1's comment (3), we have added sentences to the Discussion chapter (page 5, 3rd paragraph, 9th - page 6, 1st paragraph, 1st line "While RMnO₃ and ...and helical period.>").
4. To respond to the reviewer 2's comment (4), we have omitted the words "Nambu-Goldstone mode", and alternatively used the words "toroidal magnon mode" in the main text.
5. To respond to the reviewer 2's comment (5), we have improved the English in the revised manuscript by using a commercial proofreading service.
6. To respond to the reviewer 3's comment (1), we revised the first section ("Variation of magnetic structure in magnetic field") and second section ("Microwave absorption") in the Results chapter and newly added the Fig. 4b in the main text and Fig. S1 and S2 in the Supplementary Information.
7. Following the reviewer 3's comment (2), we newly added the absorption spectra in paraelectric (Ferri) phase to Fig. 2(c) and Fig. S2.
8. Following the reviewer 3's comment (3), we have modified the expression about the magnitude of magnetization (page 3, second paragraph, 7th-8th line "Therefore there was a finite spontaneous magnetization.>").

****Page1****

(Editor's comment 1) "Please avoid using "new, first, novel" to describe your own results."

(Response) We eliminated the word "new" following the editor's request (page 1, 12nd-13rd lines "This approach...microwave properties.").

****Page2****

(Editor's comment 2) "Our style does not permit display items in the Introduction, unless they are truly introductory."

(Response) We would like to maintain the phrase "as shown in Fig. 1a", because we believe Figure 1a is truly introductory.

(Editor's comment 3) "We are committed to ensuring clarity and avoiding ambiguity in the mathematics in our papers. Consequently, please carefully check the mathematical terms throughout your manuscript (including labels on figures and figure captions) to ensure that it conforms strictly to the following guidelines. In mathematical terms, scalar variables (e.g. x , V , χ) and constants (e.g. π , \hbar , e) should be typeset in italics, and vectors (such as r , the wave vector k , or the magnetic field vector B) should be typeset in bold without italics. In contrast, subscripts and superscripts should only be italicized if they too are variables or constants. Those that are labels (such as the 'c' in the critical temperature, T_c , the 'F' in the Fermi energy, E_F , or the 'crit' in the critical current, I_{crit}) should be typeset in roman. Please also ensure the same convention is followed in figure labels, axes, and such. Additionally, to avoid doubt, unit dimensions should be expressed using negative integers (e.g. $\text{kg m}^{-1} \text{s}^{-2}$ not kg/ms^2) or the word 'per'.

* Please ensure the same conventions are followed in the figures, so that there is consistency between figures and text."

(Response) Following the editor's comment, we corrected the usage of bold font (page 2, 1st paragraph, 19th-21st lines).

(Editor's comment 4) "Please start a new paragraph from "Here,...". The final paragraph should be a brief summary of the major results and conclusions. The results of the current study should only be discussed in this final paragraph." (page2)

(Response) Following the editor's request, we start a new paragraph from "Here,...".

****page3****

(Editor's comment 5) "Label all panels alphabetically. We do not allow to use "left,

right..." to denote panels." (page 3)

(Response) Because the left and right parts of Fig 2a are not separable, we would like not to label them individually. Alternatively we modify the text in order not to use the phrase "left panel" or "right panel" in the revised text.

(Editor's comment 6) "Vectors should be typeset in bold font without italics."

(Response) Following the editor's comment we correct the usage of font.

****page4****

(Editor's comment 7) "A specific item must be cited, please also see remarks of the annotated Supplementary Information file.

SUPPLEMENTARY INFORMATION

Please follow our style and see comments in annotated Supplementary Information file for guidance on citing the SI. At least some Supplementary Information (Figure, Table, Notes) elements need to be cited in the main paper."

(Response) We revised the sentence following the editor's comment in page 4, 1st paragraph, 7th-10th lines "When the magnetic... and 1b)".

****page7****

(Editor's comment 8) "Data availability statements and data citations policy: All Nature Communications manuscripts must include a Data availability statement at the end of the Methods section or main text (if no Methods). For more information on this policy, and a list of examples, please see <http://www.nature.com/authors/policies/data/data-availability-statements-data-citations.pdf>

- Accession codes for deposited data
- Other unique identifiers (such as DOIs and hyperlinks for any other datasets)
- At a minimum, a statement confirming that all relevant data are available from the authors
- If applicable, a statement regarding data available with restrictions
- If a dataset has a Digital Object Identifier (DOI) as its unique identifier, we strongly encourage including this in the Reference list and citing the dataset in the Data Availability Statement."

(Response) We added the Data Availability statement following the editor's request in page 7, 2nd paragraph.

****page10****

(Editor's comment 9) "A statement of competing interests should be provided."

(Response) We added the statement of competing financial interests, following the editor's request in page 10, 4th paragraphs.

****page11****

(Editor's comment 10) "Label all panels alphabetically."

(Response) We newly added a label in Fig. 1 following the editor's request.

****page12****

(Editor's comment 11) "Do not use "left, right" to denote panels."

(Response) Because the left and right parts of Fig 2a are not separable, we would like not to label them individually. Alternatively we modify the text in order not to use the phrase "left panel" or "right panel" in the revised text.

(Editor's comment 12) "Make sure that variables/constants comply with the convention as per main text."

(Response) We corrected the usage of font in Fig. 3 following to the editor's comment.

(Editor's comment 13) "The y-axis has double-labeling. Please correct it."

(Response) We corrected the double-labeling problem in the Fig. 3 and the Supplementary Fig. 2 following the editor's comment.

(Editor's comment 13) "Avoid using bounding boxes for panels."

(response) We eliminated the bounding boxes in Fig. 3 following the editor's comment.

[In Supplementary Figures]

(Editor's comment 14) "Remove the heading."

(Response) We removed the heading following the comment.

(Editor's comment 15) "make sure that variables/constants should comply with the convention as per main text."

(Response) Following the editor's comment we correct the usage of font.

(Editor's comment 16) "Supplementary Figure 1"

(Response) We changed the Figure labels in supplementary information following the editor's comment.

All the changes made are listed below:

1. We eliminated the word "new" following the editor's comment (page 1, 12nd-13rd lines "This approach...microwave properties.>").
2. We corrected the usage of bold font following the editor's comment at page 2, 1st paragraph, 19th-21st lines.
3. We divided the paragraph of the Introduction following the editor's comment (page 2, 2nd paragraph).
4. Following the editor's comment, we revised the sentence in page 3, 2nd paragraph, in order to avoid using the phrase "left panel" or "right panel".
5. We changed the font of "q₀" from italic to bold roman following the editor's comment.
6. We revised the sentence in order to specify the reference figure number of supplementary information following the editor's comment in page 4, 1st paragraph, 7th-11th lines "When the magnetic... and 1b)".
7. We added the Data Availability statement following the editor's request in page 7.
8. We added the statement of Competing financial interests following the editor's request in page 10.
9. We newly added a label in Fig. 1 following the editor's comment.
10. We revised the y-axis label in the Fig. 3 and the Supplementary Fig. 2 following the editor's comment.
11. We eliminated the bounding boxes in Fig. 3 following the editor's comment.
12. We eliminated the title of Supplementary Information and changed the Figure label "Figure S1(S2)" to "Supplementary Figure 1(2)" following the editor's comment.
13. We have tried to improve the English language using "Nature Research Editing Service" following the editor's recommendation.